

SciPost Phys. Lect. Notes 66 (2023)

# Parametric couplings in engineered quantum systems

**Anja Metelmann[1,2]**

**1** Dahlem Center for Complex Quantum Systems and Fachbereich Physik,
Freie Universität Berlin, 14195 Berlin, Germany
**2** Institute for Theory of Condensed Matter,
Karlsruhe Institute of Technology, 76131 Karlsruhe, Germany

anja.metelmann@kit.edu

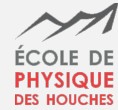

*Part of the Quantum Information Machines
Session 113 of the Les Houches School, July 2019
published in the Les Houches Lecture Notes Series*

## Abstract

Parametric couplings in engineered quantum systems are a powerful tool to control, manipulate and enhance interactions in a variety of platforms. It allows us to bring systems of different energy scales into communication with each other. This short chapter introduces the basic principles and discusses a few examples of how one can engineer parametric amplifiers with improved characteristics over conventional setups. Clearly, the selected examples are author-biased, and other interesting proposals and implementations can be found in the literature. The focus of this chapter is on parametric effects between linearly coupled harmonic oscillators, however, parametric modulation is also applicable with nonlinear couplings and anharmonic systems.

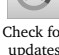

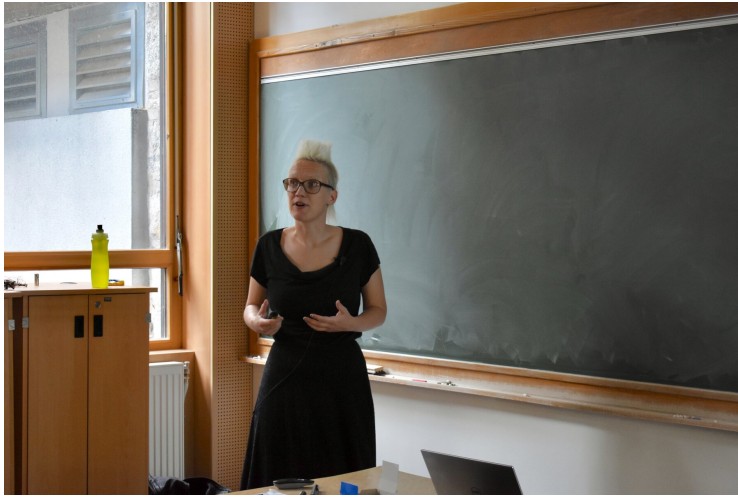



# 1 Introduction

Parametric modulation offers the exciting possibility to manipulate and control resonant interactions between engineered quantum systems. Such systems are artificial mesoscopic systems whose dynamics are governed by the laws of quantum mechanics. Taking for example cavity optomechanical platforms, the basic system consists here of a photonic mode inside a cavity coupled to a mechanical mode via radiation pressure force. The general configuration combines a low frequency mechanical mode with a high frequency photonic mode, and the interaction between them is activated by applying an external driving tone. The optomechanical Hamiltonian reads ($\hbar = 1$)

$$\hat{\mathcal{H}}_{\text{om}} = \omega_m \hat{b}^\dagger \hat{b} + \omega_c(\hat{x})\hat{a}^\dagger \hat{a} \approx \omega_m \hat{b}^\dagger \hat{b} + \left[\omega_c + g\left(\hat{b} + \hat{b}^\dagger\right)\hat{a}^\dagger \hat{a}\right], \tag{1}$$

with $\hat{a}(\hat{b})$ as the annihilation operator of the photonic field (mechanical mode) of resonant frequency $\omega_c(\omega_m)$. The mechanical motion modulates the cavity field and in general it is sufficient to consider the interaction up to linear order in the displacement $\hat{x} = (\hat{b} + \hat{b}^\dagger)x_{zpf}$ with $x_{xpf}$ as the zero point fluctuations amplitude, such that the optical shift per displacement is $g = \partial \omega_c/\partial x$. One can then perform a displacement transformation $\hat{a} = \alpha(t)e^{i\phi} + \hat{d}$ with the classical field induced by the drive $\alpha(t)$ and $\hat{d}$ denoting the deviations around the displacement. For weak coupling $g$ one can linearize the interaction leading to the Hamiltonian

$$\hat{\mathcal{H}}_{\text{om}} \approx \omega_m \hat{b}^\dagger \hat{b} + \omega_c \hat{d}^\dagger \hat{d} + G(t)\left(\hat{b} + \hat{b}^\dagger\right)\left(\hat{d}e^{-i\phi} + \hat{d}^\dagger e^{+i\phi}\right), \tag{2}$$

with the parametric modulation $G(t) = g\alpha(t)$. Applying appropriate driving tone(s) allows to explore the realm of mechanical modes in the quantum regime. For a detailed review of optomechanical systems please see Ref. [1].

Other engineered quantum systems which benefit from parametric modulation are superconducting circuit platforms. Parametric modulation forms here the basis for tunable couplers for qubits and for processing information on chip. There are abundant design choices – from a theory perspective they all are like magical loops. The basic principle can be brought down

to having superconducting rings intersected with a number of Josephson junctions. The loop is either driven with external microwave tones or the flux through the loop is modulated. An example of such a magical ring is the Josephson parametric converter (JPC) [2], a loop intersected with four Josephson junctions realizing the three wave mixing interaction

$$\hat{\mathcal{H}}_{\text{JPC}} = g_3 \left(\hat{a} + \hat{a}^\dagger\right)\left(\hat{b} + \hat{b}^\dagger\right)\left(\hat{c} + \hat{c}^\dagger\right),\tag{3}$$

with the coupling strength $g_3$ between three microwave modes $a, b, c$. A pump tone at frequency $\omega_p$ applied to the mode $c$ allows us to approximate $\hat{c} = \bar{c}e^{+i\omega_p t + i\phi}$ under a stiff pump approximation [3]. Substituting this into $\hat{\mathcal{H}}_{\text{JPC}}$ we recover the same form of the optomechanical parametric Hamiltonian in Eq.(2) with $G(t) = 2g_3\bar{c}\cos(\omega_p t + \phi)$.

Hence, taking these simple examples of engineered systems, we see that either platform generates the same linear dynamics between two oscillators. This similarity can be extended to larger networks of parametrically coupled oscillators. From a theoretical perspective this allows for the freedom to propose protocols and applications based on parametric processes which can be applied to either architecture (with benefits and limitations distinct to each platform). In the next sections we first discuss in more detail the two coupled oscillator case and then turn to the the example of designing parametric amplifiers with improved properties over conventional designs.

## 2 Engineering coherent and dissipative processes

### 2.1 Engineering coherent processes

In the former section, we have introduced two engineered system architectures which realize parametric coupling between two oscillators. Now we dive more into the details of the resulting engineered processes. We start by considering the most basic example of two harmonic oscillators with distinct frequencies $\omega_a$ and $\omega_b$. The two oscillators are coupled via a parametric modulation $\mathcal{M}(t)$ and the Hamiltonian reads ($\hbar = 1$)

$$\hat{\mathcal{H}} = \hat{\mathcal{H}}_0 + \mathcal{M}(t)\left(\hat{a} + \hat{a}^\dagger\right)\left(\hat{b} + \hat{b}^\dagger\right), \quad \hat{\mathcal{H}}_0 = \omega_a \hat{a}^\dagger \hat{a} + \omega_a \hat{b}^\dagger \hat{b},\tag{4}$$

where $\hat{a}(\hat{b})$ annihilates an excitation in oscillator $a(b)$, while $\hat{a}^\dagger(\hat{b}^\dagger)$ creates an excitation. The operators obey bosonic commutation relations, i.e., $[\hat{o}, \hat{o}^\dagger] = 1$, $(o = a, b)$. Here $\hat{\mathcal{H}}_0$ denotes the free energy of the two oscillators and the interaction between the oscillators is quadratic in creation and annihilation operators, which leads to a linear system of equations of motion for the operators. We now move into an interaction frame with respect to the free Hamiltonian $\hat{\mathcal{H}}_0$, i.e., we perform a unitary transformation of the form $\hat{\mathcal{H}}' = \hat{U}^\dagger(t)\hat{\mathcal{H}}\hat{U}(t) - \hat{\mathcal{H}}_0$ with $\hat{U}(t) = e^{-i\hat{\mathcal{H}}_0 t}$. Note that such an interaction frame is distinct from the interaction picture used in quantum mechanics [4]. In an interaction picture operators evolve with the 'boring' Hamiltonian $\hat{\mathcal{H}}_0$, while the states evolve with $\hat{V}(t) = \hat{\mathcal{H}} - \hat{\mathcal{H}}_0$. In contrast, in the interaction frame (or rotating frame) states and operators evolve with the same Hamiltonian $\hat{\mathcal{H}}'$. Thus, it is not really a picture, it simply neglects the 'boring dynamics' associated with $\hat{\mathcal{H}}_0$ after the unitary transformation has been performed. The Hamiltonian for the two oscillators in the interaction frame becomes

$$\hat{\mathcal{H}}' = \mathcal{M}(t)\left(\hat{a}\hat{b}^\dagger e^{-i(\omega_a - \omega_b)t} + \hat{a}^\dagger \hat{b}^\dagger e^{+i(\omega_a + \omega_b)t} + h.c.\right).\tag{5}$$

In this interaction frame it becomes obvious that for constant coupling, i.e., $\mathcal{M}(t) = \mathcal{M}$, the interaction processes are non-resonant and thus suppressed for $\omega_a \gg \omega_b$ and moderate coupling strength $\mathcal{M} \ll \omega_a \pm \omega_b$. In contrast, parametric modulation allows for engineering

these processes in such a way that they become resonant. The simplest single-tone parametric modulation takes the form

$$\mathcal{M}(t) = 2\lambda \cos(\omega_d t + \phi) = \lambda \left( e^{+i(\omega_d t + \phi)} + e^{-i(\omega_d t + \phi)} \right), \qquad (6)$$

where $\lambda$ denotes the driving strength, $\omega_d$ the driving frequency and $\phi$ accounts for a possible phase of the drive. By substituting this into the Hamiltonian given in Eq.(5) we find that for driving at these frequency difference of the oscillators resonant frequencies $\omega_d = \omega_a - \omega_b$ the hopping interaction becomes resonant:

$$\hat{\mathcal{H}}' = \hat{\mathcal{H}}_{\text{FC}} + \hat{\mathcal{H}}_{\text{CR}} = \lambda \left( \hat{a} \hat{b}^\dagger e^{+i\phi} + \hat{a}^\dagger \hat{b} e^{-i\phi} \right) + \hat{\mathcal{H}}_{\text{CR}}, \qquad (7)$$

where $\hat{\mathcal{H}}_{\text{CR}}$ contains the so-called counter-rotating (CR) terms which are in general neglected under a rotating wave approximation. The process in $\hat{\mathcal{H}}_{\text{FC}}$ is called frequency conversion (FC), as a photon with frequency $\omega_a$ is converted to $\omega_b$ and vice versa. Crucially, the external modulation enables a perfect way to control the interaction: $\lambda$ determines the strength of the interaction and one can as well imprint a phase $\phi$ onto the process. However, driving at the sum of the resonant frequencies of the two oscillators $\omega_d = \omega_a + \omega_b$, we obtain the parametric amplifier (PA) interaction:

$$\hat{\mathcal{H}}' = \hat{\mathcal{H}}_{\text{PA}} + \hat{\mathcal{H}}'_{\text{CR}} = \lambda \left( \hat{a}^\dagger \hat{b}^\dagger e^{+i\phi} + \hat{a} \hat{b} e^{-i\phi} \right) + \hat{\mathcal{H}}'_{\text{CR}}, \qquad (8)$$

which describes the simultaneous creation of two excitations in both oscillators, also called a two-mode squeezing interaction as one linear combination of the fields becomes squeezed. Clearly, we can as well have a two-tone parametric drive with $\mathcal{M}(t) = 2 \sum_{n=1,2} \lambda_n \cos(\omega_{d,n} t + \phi_n)$, allowing for the simultaneous realization of the frequency conversion and the parametric amplifier processes.

## 2.2 Engineering dissipative processes

So far we have considered the engineering of resonant processes between two oscillators which were coherent in nature. Now we want to turn to another process between two oscillators which we refer to as a dissipative process. The latter means that the process is mediated by a damped auxiliary system. Recall that conventional dissipation can be modeled within a system-bath theory [5], where the system of interest is coupled to a large reservoir. Within a Born-Markov approximation, i.e., assuming that the system-bath coupling is weak and that the bath is unchanged under the system-bath interaction, the reduced density matrix $\hat{\rho}_S$ of the system can be modeled as a Lindblad master equation $\frac{d}{dt}\hat{\rho}_S = -i[\hat{\mathcal{H}}_S, \hat{\rho}_S] + \mathcal{L}[\hat{z}]\hat{\rho}_S$, with the superoperator $\mathcal{L}[\hat{z}]\hat{\rho}_S = \hat{z}\hat{\rho}_S\hat{z}^\dagger - 1/2\hat{z}^\dagger\hat{z}\hat{\rho}_S - 1/2\hat{\rho}_S\hat{z}^\dagger\hat{z}$ and $\hat{z}$ as the so-called jump-operator. The first term denotes a possible coherent evolution of the system under $\hat{\mathcal{H}}_S$, while the second term describes the incoherent dynamics induced by the bath. In a conventional setting the incoherent dynamics is detrimental to the system, e.g. quantum states decohere and information is lost into the bath. In contrast, within the framework of reservoir engineering one uses dissipation to one's advantage. Here one specifically designs the environment to achieve desired dynamics or to cool a system to specific states [6]. We are interested in using such dissipation engineering to generate a dissipative process between two oscillators. Therefor we extend our two oscillator setup. Instead of coupling them directly, we couple them indirectly via an auxiliary mode $c$. Taking the example of hopping interactions we start from the system-bath (SB) Hamiltonian

$$\hat{\mathcal{H}}_{\text{SB}} = \Delta \hat{c}^\dagger \hat{c} + \lambda \left[ \hat{c}^\dagger \left( \hat{a} + \hat{b} \right) + h.c. \right] + \hat{\mathcal{H}}_{c,\text{diss}}, \qquad (9)$$

with $\hat{\mathcal{H}}_{c,\text{diss}}$ denoting the coupling of the auxiliary mode to a (zero-temperature) bath with rate $\kappa$, and $\Delta$ accounting for a detuning of the $c$-mode. Assuming that the $c$-mode is strongly

damped and weakly coupled to the main oscillator pair $a$ and $b$, we can adiabatically eliminate it [7], and obtain the master equation

$$\frac{d}{dt}\hat{\rho} = -i\Lambda\left[\left(\hat{a}^{\dagger} + \hat{b}^{\dagger}\right)\left(\hat{a} + \hat{b}\right), \hat{\rho}\right] + \Gamma\mathcal{L}\left[\hat{a} + \hat{b}\right]\hat{\rho},$$

$$\text{with} \quad \Lambda = \frac{\Delta\lambda^2}{\Delta^2 + \frac{\kappa^2}{4}}, \quad \text{and} \quad \Gamma = \frac{\kappa\lambda^2}{\Delta^2 + \frac{\kappa^2}{4}}, \tag{10}$$

denoting effective coherent and dissipative coupling strength $\Lambda$ and $\Gamma$ respectively. The effective coherent dynamics with strength $\Lambda$ are only present for finite detuning $\Delta \neq 0$. They induce frequency shifts and coherent hopping between the oscillators. The dissipative process on the other hand can be reformulated as

$$\mathcal{L}\left[\hat{a} + \hat{b}\right]\hat{\rho} = \mathcal{L}\left[\hat{a}\right]\hat{\rho} + \mathcal{L}\left[\hat{b}\right]\hat{\rho} + \left[\hat{a}\hat{\rho}\hat{b}^{\dagger} - \frac{1}{2}\left\{\hat{b}^{\dagger}\hat{a}, \hat{\rho}\right\} + h.c.\right]. \tag{11}$$

The first two terms describe local damping of the mode $a$ and $b$ respectively, while the remaining terms correspond to a swapping interaction between the two oscillators, we refer to this as dissipative hopping as it is mediated by the reservoir (the damped $c$-mode). Due to this indirect hopping the phase associated with the dissipative hopping process differs from the coherent hopping process.

## 2.3 Breaking the symmetry of reciprocity

We have learned how to engineer coherent and dissipative processes via parametric modulation. In what follows, we are going to discuss how the interplay of a coherent and a dissipative process allows for the breaking of the symmetry of reciprocity. The basic method can be illustrated as follows [8]. Starting out with two independent systems A and B which interact coherently (and bi-directionally) via a Hamiltonian $H_{\text{coh}} = \lambda/2 \, \hat{A}\hat{B} + h.c.$ where $\hat{A}(\hat{B})$ is a system A(B) operator, and $\lambda$ denotes the coupling strength. To achieve nonreciprocity, we assume both systems have been jointly coupled to the same engineered reservoir. In the Markovian limit, the engineered reservoir can be adiabatically eliminated and the full dissipative dynamics of the systems is then described by the Lindblad master equation

$$\frac{d}{dt}\hat{\rho} = -i\left[\hat{\mathcal{H}}_{\text{coh}}, \hat{\rho}\right] + \Gamma\mathcal{L}\left[\hat{A} + \eta e^{i\varphi}\hat{B}^{\dagger}\right]. \tag{12}$$

Directionality is now achieved by balancing the dissipative and the coherent interaction, i.e., by setting $\lambda = \eta\Gamma$ and $\varphi = \pm\pi/2$. The sign of the phase $\varphi$ determines the 'direction' of the interaction: for $\varphi = -\pi/2$ system B is influenced by the evolution of system A, but system A evolves independently of system B or vice versa for $\varphi = \pi/2$. Crucially, to realize nonreciprocity the Markovian limit is not a necessary condition, it only affects the directionality bandwidth, i.e., non-Markovian effects decrease the frequency range over which a system is uni-directional [9]. For example, in [10] it was possible to observe strong nonreciprocity in an optomechanical array, despite having a non-Markovian reservoir.

Unidirectionality of information transport is of paramount importance for reading out a quantum system without perturbing the signal source. For nonreciprocal photon transmission, approaches based on refractive-index modulation [11], optomechanical interaction [12], and interfering parametric processes [13] have been considered. The directionality protocol described here can serve as a recipe to realize directional photonic devices.

# 3 Quantum-limited parametric amplifiers

With the advances in quantum technologies over the last years, especially in superconducting circuitry, it has become necessary to detect and process signals containing only a few photons. To enable the detection of such weak signals with high efficiency, new amplifiers have been developed which have the ability to operate efficiently in the quantum regime. The aim is here to amplify the weak signal without noise contamination beyond the so-called quantum limit, i.e., $P_{\text{out}} = \mathcal{G}(P_{\text{in}} + \bar{n}_{\text{add}})$ with the output signal $P_{\text{out}}$ containing the input signal $P_{\text{in}}$, enhanced by the gain factor $\mathcal{G} > 1$. The minimal added noise $\bar{n}_{\text{add}}$ is determined by the operation mode of the amplifier [14], which can be either phase-insensitive ($\bar{n}_{\text{add}} = 1/2$), i.e., amplifying both quadratures of the light field, or phase-sensitive ($\bar{n}_{\text{add}} = 0$), amplifying one quadrature of the field. In what follows we are going to focus on cavity-based amplifiers, although extended structures such as traveling-wave amplifiers exist too [15]. A promising route to design quantum amplifiers is based on parametric modulation of coupled modes and the default interactions are $\hat{\mathcal{H}}_{\text{DPA}} = \lambda \hat{a}^\dagger \hat{a}^\dagger + \lambda^* \hat{a}\hat{a}$, describing degenerate parametric amplification (DPA) also called single mode squeezing, and $\hat{\mathcal{H}}_{\text{PA}} = \lambda \hat{a}^\dagger \hat{b}^\dagger + \lambda^* \hat{a}\hat{b}$, corresponding to two mode squeezing of the so-called idler and signal modes $a$ and $b$. Both interactions enable quantum limited amplification and there are numerous ways to realize them, for more details please see for example [16]. However, cavity-based amplifiers have some intrinsic shortcomings, as they amplify signals over a bandwidth $\Delta\omega$ which is inversely proportional to the amplitude gain, i.e, $\Delta\omega \propto 1/\sqrt{\mathcal{G}}$. This means that by enhancing the gain, the bandwidth of the gain profile shrinks. The latter is quantified by a so-called gain-bandwidth product, which is constant for the upper examples of amplifiers. The origin for this is that the mechanism of amplification involves approaching an instability (or 'lasing' threshold), allowing for growth but effectively introducing anti-damping which decreases the linewidth of the resonance. Another disadvantage is that the back-reflected signal is amplified as well. Moreover, these amplifiers are reciprocal and thus amplify in both directions, which makes it not possible to connect them directly to the signal source, as amplified noise arising from the measurement chain could disturb the source significantly. In the following sections we will discuss how we can design quantum-limited amplifiers overcoming these shortcomings.

## 3.1 Phase-insensitive amplifier without gain-bandwidth limit

We start with a phase-insensitive amplifier without a gain bandwidth-product. We couple the signal and idler mode not directly, but via an auxiliary mode $c$, thus we call it the dissipative amplifier (DA). The basic interaction Hamiltonian becomes

$$\hat{H}_{\text{DA}} = g\hat{c}^\dagger\left(\hat{d}_1 + \eta\hat{d}_2^\dagger\right) + h.c., \tag{13}$$

where $\hat{c}^\dagger$ denotes the creation operator of the auxiliary mode, while the operators $\hat{d}_{1,2}$ destroy an excitation in the signal and idler mode respectively. The mode $d_{1,2}$ is coupled with strength $g(g\eta)$ to the auxiliary system, with $\eta$ as a dimensionless factor accounting for an asymmetry in the couplings. Here we work in an interaction frame with respect to the free Hamiltonian and already applied a rotating wave approximation. Taking what we have learned above about parametric modulation, we assumed that the system is driven with two pump tones, one at the frequency difference of the signal and auxiliary mode, and one at the sum of the idler and auxiliary mode.

To illustrate the origin of the amplification process, we adiabatically eliminate the auxiliary mode $c$ (assuming it is strongly damped with rate $\kappa_c$), and obtain the effective dissipative

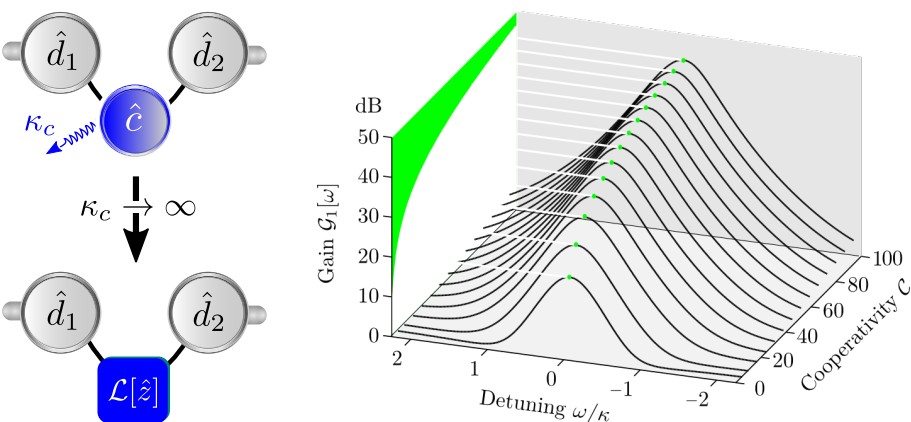

Figure 1: Dissipative amplification process between two modes $d_{1,2}$ realized via a coupling to an overdamped auxiliary mode (left). The frequency dependent gain shows no reduced bandwidth when the gain is enhanced (right).

process between the idler and signal modes ($\eta = 1$ for simplicity)

$$\Gamma \mathcal{L}\big[\hat{d}_1 + \hat{d}_2^\dagger\big]\hat{\rho} = \Gamma \mathcal{L}\big[\hat{d}_1\big]\hat{\rho} + \Gamma \mathcal{L}\big[\hat{d}_2^\dagger\big]\hat{\rho} + \Gamma\left[\hat{d}_1\hat{\rho}\hat{d}_2 - \frac{1}{2}\big\{\hat{d}_2\hat{d}_1, \hat{\rho}\big\} + h.c.\right], \qquad (14)$$

with $\Gamma = \frac{4g^2}{\kappa_c}$. The expansion of the non-local superoperator in the second step explains what processes are mediated by the auxiliary mode. The first term denotes damping of the signal mode with rate $\Gamma$, while the second term denotes anti-damping of the idler mode. The remaining terms realize the effective coupling between the two modes, which by itself just realizes a coherent rotation of the modes. The combined process can be understood as dissipative phase-insensitive amplification, and the qualitatively equivalent coherent process would be $\hat{\mathcal{H}}_{PA}$ [17].

To quantify the properties of the amplifier we work with the full system Hamiltonian in Eq.(13) and assume that all modes are coupled to external waveguides with rate $\kappa$. Utilizing input-output theory [18] and we obtain the power gain ($\eta = 1, \omega' = 2\omega/\kappa$)

$$\mathcal{G}[\omega] = \frac{\big[\sqrt{\mathcal{G}_0} - \omega'^2\big]^2 + \omega'^2\big[1 + \omega'^2\big]^2}{\big[1 + \omega'^2\big]^3}, \quad \mathcal{G}_0 = (2\mathcal{C} - 1)^2, \qquad (15)$$

for a signal injected into the mode $d_1$ (details of the calculation can be found in [19]). Crucially, the gain can be made arbitrarily large by increasing the cooperativity $\mathcal{C} = \Gamma/\kappa$ with no corresponding reduction of bandwidth (which remains $\sim \kappa$), see Fig. 1. Thus, there is no fundamental limitation on the gain-bandwidth product in this system. Moreover, including the asymmetry parameter $\eta$ the bandwidth increases for $\eta < 1$ until the onset of visible mode-splitting at $\eta = \sqrt{1 - 1/\mathcal{C}}$. In addition, although the mode-space has increased due to the mode $c$ the amplifier is still quantum limited. In the large gain limit, the added noise yields $\bar{n}_{\text{add}} \approx 1/2 + \bar{n}_{d_2}^T + 2(1 + \bar{n}_{d_2}^T + \bar{n}_c^T)/\sqrt{\mathcal{G}_0}$, with $\bar{n}_o^T$ as the thermal occupation of the bath of mode $o = c, d_2$. Thus, if mode $d_2$ is driven purely by vacuum noise, then in the large-gain limit the amplifier approaches the standard quantum limit of a phase-preserving amplifier.

## 3.2 Phase-sensitive amplifier without gain-bandwidth limit

It is also possible to design a phase-sensitive amplifier without a gain-bandwidth product. We have learned above that parametric modulation of a pair of linearly coupled cavity modes

results in two basic interactions: frequency conversion for modulating at the frequency difference of the mode pair, or parametric amplification if one drives at the sum of their frequencies. We utilize both of these interactions simultaneously, i.e., we start from the Hamiltonian involving the gain (G) and conversion (C) process

$$\hat{H}_{\text{GC}} = (G_1 + G_2)\hat{X}_1\hat{X}_2 + (G_2 - G_1)\hat{P}_1\hat{P}_2, \tag{16}$$

in the quadrature basis $\hat{X}_n = (\hat{d}_n + \hat{d}_n^\dagger)/\sqrt{2}$ and $\hat{P}_n = -i(\hat{d}_n - \hat{d}_n^\dagger)/\sqrt{2}$ ($n \in 1, 2$). The coupling coefficients $G_n$ containing the amplitude of two external modulation tones. We can couple both modes to external waveguides with rate $\kappa$ and determine the scattering matrix

$$\begin{pmatrix} \hat{X}_{1,\text{out}}[\omega] \\ \hat{P}_{2,\text{out}}[\omega] \end{pmatrix} = \mathbf{s}[\omega] \begin{pmatrix} \hat{X}_{1,\text{in}}[\omega] \\ \hat{P}_{2,\text{in}}[\omega] \end{pmatrix}, \quad \mathbf{s}[\omega] = \begin{pmatrix} \mathcal{R}[\omega] & \mathcal{T}_-[\omega] \\ \mathcal{T}_+[\omega] & \mathcal{R}[\omega] \end{pmatrix}, \tag{17}$$

with the reflection and the transmission scattering amplitudes ($\omega' = 2\omega/\kappa$)

$$\mathcal{R}[\omega] = \frac{\Delta\mathcal{C} + [1 + \omega'^2]}{\Delta\mathcal{C} - [1 - i\omega']^2}, \quad \mathcal{T}_\pm[\omega] = \frac{\sqrt{\mathcal{G}_\pm}[1 - \Delta\mathcal{C}]}{\Delta\mathcal{C} - [1 - i\omega']^2}, \quad \sqrt{\mathcal{G}_\pm} = \frac{2[\sqrt{\mathcal{C}_1} \pm \sqrt{\mathcal{C}_2}]}{1 - \Delta\mathcal{C}}. \tag{18}$$

The gain-bandwidth-limit free regime is obtained for $-1 \leq \Delta\mathcal{C} \leq 0$. $\mathcal{T}_+[\omega]$ corresponds to the amplitude gain, where the output of the $P_2$-quadrature contains the amplified $X_1$-quadrature. Note, the amplifier is reciprocal, i.e., we obtain the same scattering matrix for the $X_2$- and $P_1$-quadratures. The amplification is quantum limited as there is no gain in reflection, i.e., we have $\bar{n}_{\text{add}} = |\mathcal{R}[0]|^2/2\mathcal{G}_+ \to 0$ for large gain. Thus, we have amplification in transmission involving as well a frequency conversion process. The bandwidth of the gain becomes $\Delta\omega = \sqrt{\sqrt{2[\Delta\mathcal{C}^2 + 1]} - [\Delta\mathcal{C} + 1]}\,\kappa$, which takes its maximal value of $\sqrt{2}\kappa$ for $\Delta\mathcal{C} = -1$, while it would vanish for $\Delta\mathcal{C} \to 1$ (the standard amplification regime). Besides the maximal bandwidth, the tuning of the system to $\Delta\mathcal{C} = -1$ has even more advantages. The reflection vanishes on resonance $\mathcal{R}[0] = 0$, making the system perfectly impedance matched. Moreover, we have a true phase-sensitive amplifier, in the sense that the orthogonal quadrature gets squeezed below the shot-noise value. However, the gain-independence of the amplification bandwidth does not translate to the squeezing bandwidth, but the bandwidth is larger than for a single-mode setup, i.e., for the same gain value an enhancement of the bandwidth by the factor $\mathcal{G}^{1/4}/\sqrt{2}$ is obtained. Thus, under optimal tuning conditions one can have a two-mode phase-sensitive amplifier that is ideal with respect to a number of metrics: it has distinct input and output ports, no reflection gain, is quantum-limited and it does not suffer from a gain-bandwidth limit [20].

### 3.3 Nonreciprocal amplifiers without gain-bandwidth limit

Next, we are going to briefly discuss how one can engineer a nonreciprocal amplifier without a gain-bandwidth product by applying the directionality recipe discussed in Sec.2. We focus on the coherent phase-sensitive amplifier discussed above, i.e., described by the Hamiltonian in Eq. 16, which is straightforwardly rendered nonreciprocal by combining it with its dissipative counterpart [8]:

$$\frac{d}{dt}\hat{\rho} = -iG[\hat{X}_1\hat{X}_2, \hat{\rho}] + \Gamma\mathcal{L}[\hat{X} - i\hat{X}_2], \tag{19}$$

for simplicity we focus here on the QND-case $G_1 = G_2 \equiv G/2$, where the dissipative process preserves the QND structure of the coherent Hamiltonian. The master equation is valid in the Markovian regime, i.e., where the engineered reservoir is already adiabatically eliminated. To generate the required dissipative process, we take the engineered reservoir to be a damped

auxiliary mode $c$ with $H_{\text{SB}} = \sqrt{\Gamma\kappa_c/2}(\hat{X}_1\hat{P}_c + \hat{X}_2\hat{X}_c)$. Using input-output theory and applying the directionality condition $G = \Gamma$, we obtain the gain and reverse gain (in the non-Markovian regime)

$$\mathcal{G}_{P_2 X_1}[\omega] = \frac{\mathcal{G}_0\left[1 + \frac{\omega^2}{\gamma^2}\right]}{\left[1 + \frac{4\omega^2}{\kappa_c^2}\right]\left[1 + \frac{4\omega^2}{\kappa^2}\right]^2}, \qquad \bar{\mathcal{G}}_{P_1 X_2}[\omega] = \mathcal{G}[\omega]\frac{\frac{\omega^2}{\kappa_c^2}}{\left[1 + \frac{\omega^2}{\kappa_c^2}\right]}, \qquad \mathcal{G}_0 = \frac{64\Gamma^2}{\kappa^2}, \quad (20)$$

the $P_2$-quadrature becomes the amplified copy of $X_1$-quadrature as before, but in contrast to the reciprocal case, the reverse amplification process of the $P_1$-quadrature is suppressed. On resonance, i.e., for $\omega = 0$, the reverse gain vanishes independently of the value of $\kappa_c$ and therewith independently of the Markovian limit. However, we clearly see that the Markovian limit is preferable, as for $\kappa_c \to \infty$ the reverse gain vanishes over a wide frequency regime. Thus, deviations from the Markovian limit impact the bandwidth over which directionality is observed.

Furthermore, a nonreciprocal phase-insensitive amplifier can be accomplished as well by combining the dissipative amplification process in Eq. (14) with its coherent counterpart $\hat{\mathcal{H}}_{\text{PA}}$. Such an amplifier has been introduced in the framework of a graph-theoretical approach [21], and has been successfully implemented in superconducting circuit architectures [22,23]. However, the additional coherent interaction effectively introduces anti-damping, thus the advantage of having no gain-bandwidth limit is lost. It turns out, by using a single engineered reservoir, either phase-sensitivity or no gain-bandwidth limitation can be achieved in a resonant directional amplifier (but not both properties simultaneously). However, one can achieve both of these desirable conditions in a design that utilizes two engineered reservoirs, e.g. the master equation

$$\frac{d}{dt}\hat{\rho} = -iG\left[\hat{d}_1^\dagger\hat{d}_2^\dagger + \hat{d}_1\hat{d}_2, \hat{\rho}\right] + \Gamma_1\mathcal{L}\left[\hat{d}_1^\dagger - i\hat{d}_2\right] + \Gamma_2\mathcal{L}\left[\hat{d}_1 - i\hat{d}_2^\dagger\right], \qquad (21)$$

describes a directional quantum amplifier that is both phase-preserving, and which does not suffer from a gain-bandwidth constraint, for further details please see Ref. [24].

# 4 Conclusion

Parametric modulation enables a high control over interactions among coupled subsystems, and many applications in quantum information science are based on it, e.g., entanglement generation, state preparation, computational gates or read-out. This chapter provided here a very brief introduction, starting out from parametric modulated couplings among a few harmonic oscillators to engineer coherent and dissipative processes. Clearly this was just the top of the iceberg, and hopefully encourages the reader to further explore the realm of parametric effects in engineered quantum systems.

# Acknowledgements

I would like to thank Ioan Pop, Benjamin Huard and Michel Devoret for their invitation to give a colloquium at the Les Houches summer school Quantum Information Machines in 2019 and the chance to contribute to these lecture notes. I also would like to thank Archana Kamal and Christian Arenz for comments on the manuscript.

**Funding information** The author acknowledges funding by the Deutsche Forschungsgemeinschaft through the Emmy Noether program (Grant No. ME 4863/1-1).

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
