# Peer review of "Parametric Couplings in Engineered Quantum Systems"

_SciPost Physics Lecture Notes, doi:SciPost Phys. Lect. Notes 66 (2023)_

## Round 1 · Referee Report · Anonymous (Referee 1) · 2022-9-12

Strengths

1- Engineering parametric couplings is an immensely powerful toolkit in many hardware platforms, ranging from the activation of two-qubit gates in superconducting quantum circuits with large on-off ration, to bridging the energy gap between low and high energy modes in quantum optomechanical systems. The manuscript gives a strong introduction into the basic principles of this field.
2- Parametric amplifiers are of utterly importance for the readout of quantum systems, not only in quantum information processing schemes but also in quantum sensing applications. Significantly improving the performance of established design concepts simply by applying additional microwave tones at the right frequencies and amplitudes will help to elevate this technology to become as standard as commercial high-electron-mobility (HEMT) amplifiers. The manuscript gives a complete overview on the current limitations of these amplifiers and how to overcome these shortcomings by introducing minimal changes to the operational mode.
3- In general, the manuscript is written very educationally
4- The presented ideas are new and very exciting for the field

Weaknesses

1- The only "weakness" I found is a small lack in headlines, which would give the manuscript even more structure

Report

The manuscript is very well written and meets the Journal's acceptance criteria. The discussed topic is new, exciting for the field, and will spark the creativity of scientists to develop the next generation of quantum-limited parametric amplifiers. In addition, combining different hardware platforms in a hybrid architecture to benefit from the strength of the individual systems becomes possible only if the energy gap between the subsystems can be bridged. Parametric modulation shows a clear way towards the successful implementation, and the manuscript gives an educational introduction into this field.

Requested changes

1- I would make the notation in the equations a bit more uniform. For instance, I would use "d1" and "d2" as the annihilation operators for the two linear modes throughout the whole manuscript. The same holds for the linear coupling rates, which I would label with "g" or "lambda", but not both in different sections.
2- I am suggesting to give the meaning behind the abbreviations "DA", "PA", and "DPA" in the main text.
3- I am suggesting to use round brackets to indicate the dependence of a system parameter on a variable like the drive frequency. Example: G(w) instead of G[w]. Since most regular brackets are also "[]", this choice significantly reduces the probability of confusion arising from the notation in my opinion.
4- I am suggesting to introduce some subsection headlines to give the manuscript a bit more structure. For instance, I would divide section 2 into three subsections:
a) Engineering coherent interactions
b) Dissipation engineering
c) Breaking the symmetry of reciprocity
I am suggesting a similar approach for section 3, since there are several different concepts introduced.
1) Beating the gain-bandwidth product
a) Phase-insensitive amplifiers
b) Phase-sensitive amplifiers
2) Nonreciprocal amplifiers
5- I am suggesting to introduce some concepts a bit earlier in the text. For example: The scattering matrix could be introduced earlier, to give a definition for the transmitted gain G(w) in the case of the phase-insensitive amplifier (Eq. 15)
6- In the same spirit, I am suggesting to use a more general treatment of concept from the beginning, including the relative phases and amplitudes of the individual subsystems 1 and 2 (or A and B). For instance, the Hamiltonian in Eq. 9 holds only for a particular choice of the relative phase and relative amplitude (phi = 0, eta = 1). Personally, I think it would not increase the complexity too much to account for these two additional parameters already from the start and use a notation similar to the definition given in Eq. 12.

Attachment

---

## Round 2 · Author Response

We like to thank the referee for their positive feedback and comments.
We have implemented some of the suggested changes.

---

## Round 2 · List of Changes

• headlines added to give the manuscript more structure
  • abbreviations are now defined in text

---

## Editorial Decision

published